# Thermobarometry of the Rajmahal Continental Flood Basalts and Their Primary Magmas: Implications for the Magmatic Plumbing System

**Nilanjan Chatterjee** [1,*] and **Naresh C. Ghose** [2,†]

1 Department of Earth, Atmospheric and Planetary Sciences, Massachusetts Institute of Technology, Cambridge, MA 02139, USA

2 Department of Geology, Patna University, Patna 800005, India

* Correspondence: nchat@mit.edu; Tel.: +1-617-253-1995

† Current address: G/608, Raheja Residency, Koramangala, Bangalore 560034, India.

**Abstract:** The Late Aptian Rajmahal Traps originated through Kerguelen-Plume-related volcanism at the eastern margin of the Indian Shield. Clinopyroxene and whole-rock thermobarometry reveals that the Rajmahal magmas crystallized at P-T conditions of $\leq$~5 kbar/~1100–1200 °C. These pressures correspond to upper crustal depths ($\leq$~19 km). Modeling shows that the Rajmahal primary magmas were last in equilibrium with mantle at P-T conditions of ~9 kbar/~1280 °C. The corresponding depths (~33 km) are consistent with gravity data that indicate a high-density layer at lower crustal depths below an upwarped Moho. Thus, the high-density layer probably represents anomalous mantle. It is likely that the mantle-derived magmas accumulated below the upwarped Moho and were subsequently transported via trans-crustal faults/fractures to the upper crust where they evolved by fractional crystallization in small staging chambers before eruption. In the lower part of the Rajmahal plumbing system, buoyant melts from the Kerguelen Plume may have moved laterally and upward along the base of the lithosphere to accumulate and erode the eastern Indian lithospheric root. The Rajmahal plumbing system was probably shaped by tectonic forces related to the breakup of Gondwana.

**Keywords:** flood basalt; Rajmahal Traps; thermobarometry; primary magma; magma plumbing system

## 1. Introduction

Flood basalt eruptions have important environmental effects, and magma volumes and eruption rates of flood basalts are directly related to the structure of the plumbing system through which mantle-derived magmas reach the Earth's surface [1–3]. The structure of the plumbing system in continental flood basalt provinces may depend on the regional tectonics and proximity to a mantle plume [4,5]. Exposures of dikes, sills and other magma bodies, and geophysical surveys shed light on the structure of the plumbing system [6–9]. The plumbing system of plume-related continental large igneous provinces (LIP) can be divided into four parts corresponding to four depth levels [3]. These are the asthenospheric mantle level where melt is generated (level 1), the lithospheric mantle level through which magmas ascend and accumulate below the Moho (level 2), the crustal level where magma is transported via dikes and sills into and out of small staging chambers where magma differentiation occurs (level 3), and the surface level where volcanoes develop (level 4).

An LIP related to the activity of the Kerguelen Plume has been recognized in eastern India, southwestern Australia (Bunbury), the eastern Antarctic margin, the Kerguelen Plateau, the Ninetyeast Ridge, the Broken Ridge, and the Naturaliste Plateau [10–15] (Figure 1). The oldest basalts (~132 Ma) erupted in southwestern Australia. However, there are some reports of even older Kerguelen-Plume-related volcanic activity at ~145–130 Ma in the eastern Himalayas [16]. The eastern Indian basalts and most of the basalts of the Kerguelen Plateau erupted during

the peak of Kerguelen Plume activity at ~120–95 Ma [15]. In eastern India, the Late Aptian (~118–114 Ma) Rajmahal–Bengal Basin–Sylhet Traps (RBST) and the associated dolerite dikes and alkalic-carbonatitic-ultramafic intrusives comprise a large flood basalt province with an area of ~1 million km$^2$ [17–26]. The Rajmahal and Sylhet Traps are located in the western and eastern parts of the province (Figure 2), and drill-core data indicate that the two are continuous under the Gangetic alluvium of the Bengal Basin [27]. Dolerite dikes with NW to NNW-trend and similar age (118–109 Ma) intrude the Precambrian basement to the southwest and west of the Rajmahal outcrop [28] (Figure 2). Geochemical, isotopic, and textural studies of the RBST basalts have provided clues to the processes of fractional crystallization and crustal contamination in their origin [10,19,21,23,25,26,29–32]. However, the crystallization and mantle melting conditions of the basalts are yet to be established, and the structure of the pathways through which the basalts erupted is poorly understood. Although geophysical surveys have helped to clarify subsurface structures [33,34], and two- and three-dimensional modeling of gravity data have identified possible magmatic underplating below the lower crust [35,36], the structure of the magma plumbing system in the upper mantle and crust remains unclear. Thermobarometry based on the chemical composition of minerals and whole rock provides the temperatures and depths of magma equilibration, which may be used to complement geophysical studies to understand the structure of the magma plumbing system. This study aims to understand the structure of the Rajmahal plumbing system through thermobarometry based on clinopyroxene compositions analyzed here and whole-rock compositions available from the literature. The P-T conditions of crystallization obtained from thermobarometry are then used to backtrack along the fractionation paths of the basalts and calculate compositions of primary magmas and their equilibration depths.

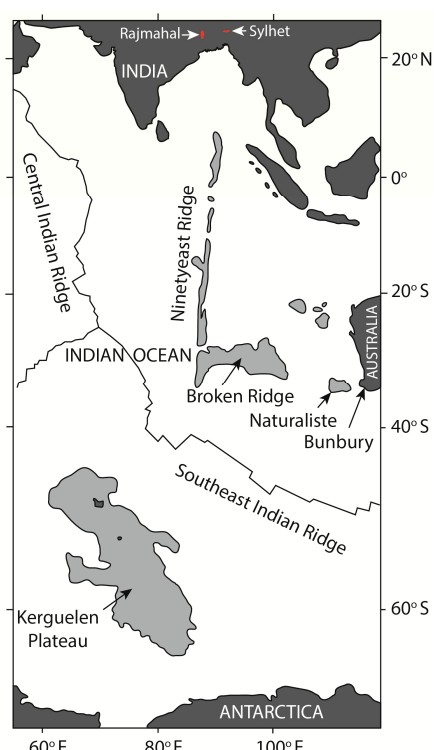

**Figure 1.** Location of the different parts of Kerguelen LIP. The continental landmasses are shown in black and the submarine plateaus and ridges are shown in grey.

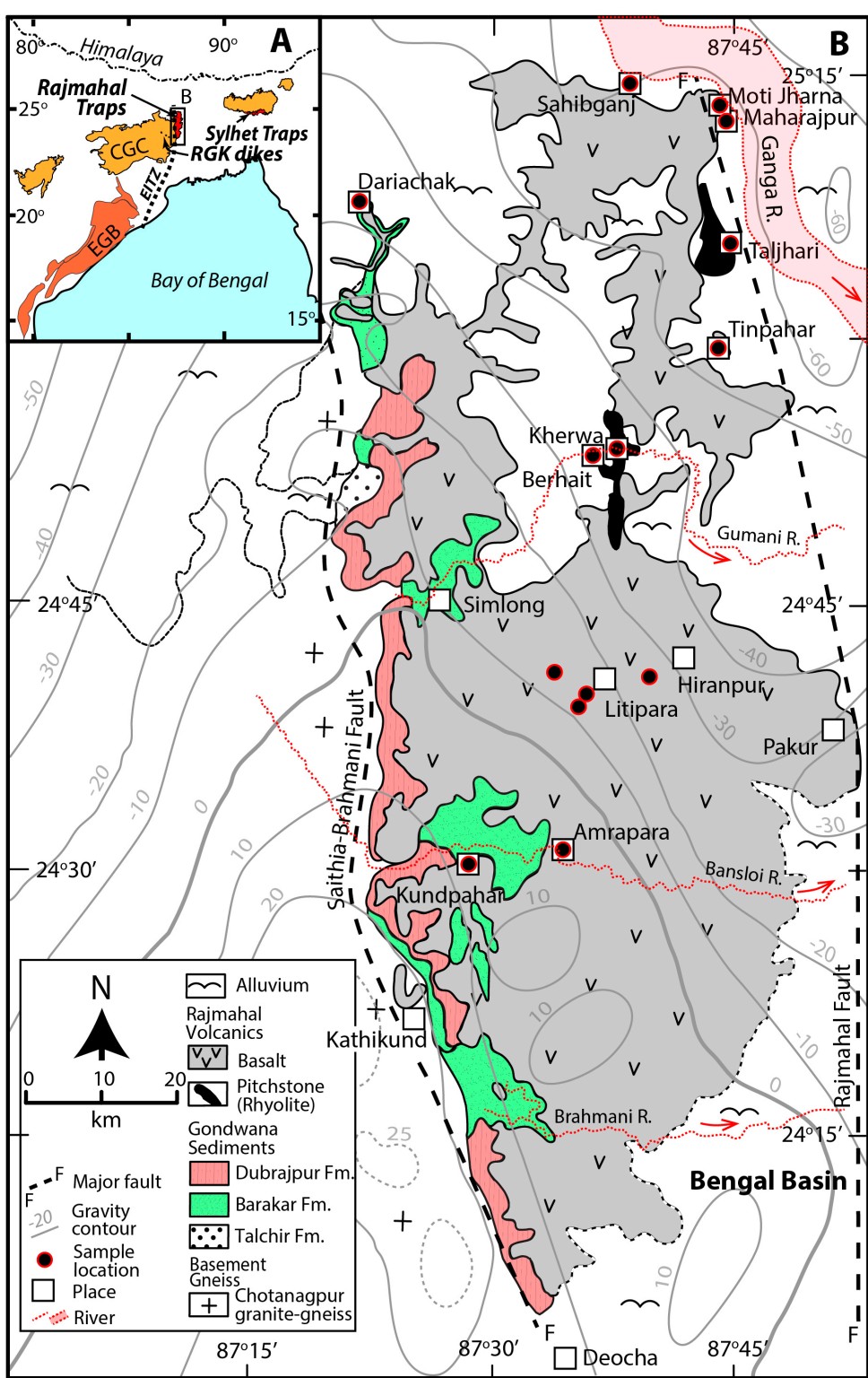

**Figure 2.** (**A**) Location of the Rajmahal Traps, Sylhet Traps, and Raniganj-Giridih-Koderma (RGK) dikes on the Indian Shield. CGC—Chotanagpur Gneiss Complex, EGB—Eastern Ghats belt, EITZ—Eastern Indian Tectonic Zone. (**B**) Detailed geological map of the Rajmahal Traps (after [32]). The Bouguer gravity contours (−60 to +25 mGal) are after [36].

## 2. Geological and Geochemical Background

The Rajmahal Traps (area: ~4300 km$^2$) are exposed along the north–south-oriented Rajmahal Hills in the northern part of the eastern margin of the Proterozoic Chotanagpur

Gneissic Complex (CGC) (Figure 2). The CGC forms the basement of the Gondwana Supergroup, the uppermost part of which contains the Rajmahal basalts [37,38]. The Rajmahal flows are sub-horizontal with the dip increasing to ~5° eastward at the eastern flank of the hills, and the total thickness of the flows increases from ~230 m in surface exposure to >332 m subsurface in the Bengal Basin [27,35]. The thickness of individual flows varies from <1 m to 85 m [39–41]. Thin beds of shale, black shale, mudstone, siltstone, cross-bedded sandstone, and oolite associated with bentonite lenses and volcaniclastic rocks occur between lava flows in the lower one-third of the volcanic sequence, indicating sub-aqueous eruptions during the early phase of volcanism [32,41–43]. The upper part of the sequence is devoid of sedimentary and volcaniclastic rocks, and consists of subaerially erupted lava flows [23,32]. The volcanics are dominated by tholeiitic basalt and basaltic andesite with minor trachyandesite, andesite, dacite and rhyolite, and rare orthopyroxene-bearing basalt and andesite [21,23,32]. The tholeiites have been classified into two geochemical groups [10] (Figure 3). The Group I samples have higher Ti/Zr ratios and lower Zr/Y ratios than the Group II samples. According to the total alkali-silica classification (TAS diagram, [44]), the Group I samples are basalt and basaltic andesite, whereas the Group II samples are all basaltic andesite. Isotopic and trace element data suggest that the Group I samples are uncontaminated, whereas the Group II samples with incompatible element patterns showing large-ion-lithophile element (LILE) enrichment and negative Nb-Ta anomalies are variably contaminated with the continental crust (Figure 3c) [10,18,21,23,26,31,32]. The chemical distinctions between the two groups of basalt also correlate with their textures, the early erupting Group I basalts in the lower part of the sequence being phyric with plagioclase and/or clinopyroxene phenocrysts, and the late erupting Group II basalts in the upper part of the sequence being mostly aphyric [23].

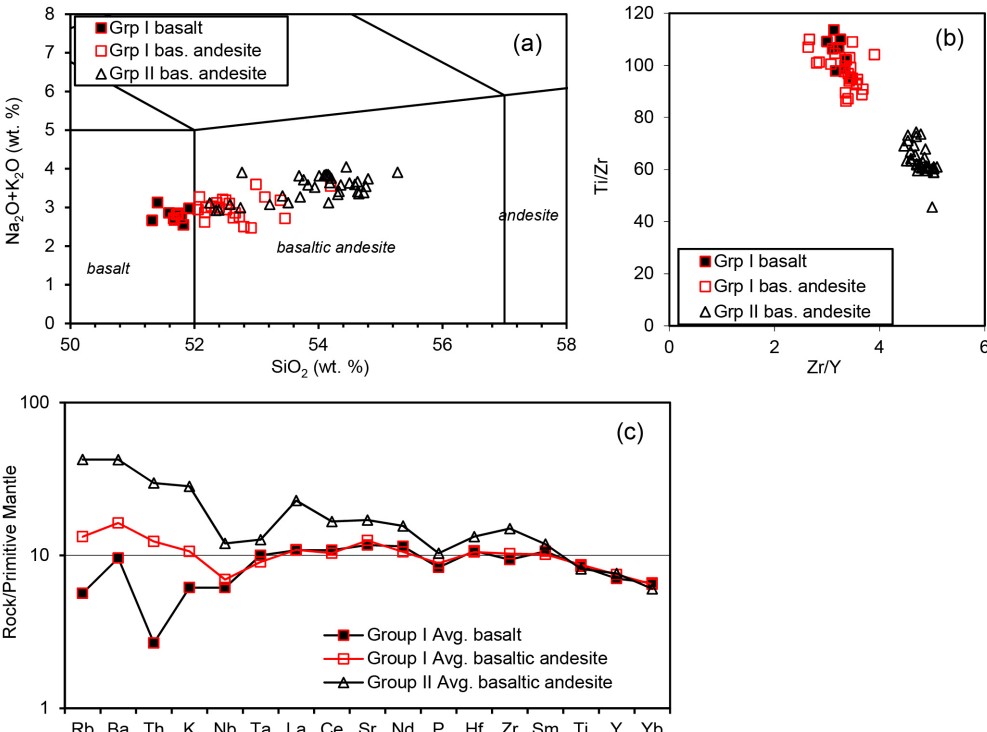

**Figure 3.** Bulk chemical compositions of the Rajmahal basalts and basaltic andesites (data from [10,21]). (**a**) Total alkali versus SiO$_2$ after [44]. (**b**) Ti/Zr versus Zr/Y. (**c**) Primitive mantle-normalized [45] incompatible elements.

The eastern margin of the CGC is characterized by north-trending, east-dipping asymmetric folds that formed during high-grade metamorphism and deformation along the north–south-oriented, mid-Neoproterozoic Eastern Indian Tectonic Zone, a major linear

orogen that extends from eastern India through Kerguelen Plateau to eastern Antarctica [46–48]. The eastern and western margins of the Rajmahal outcrop are bounded by two major N–S-oriented faults, the Rajmahal and Saithia–Brahmani faults (Figure 2), and the basement near the western margin of the Bengal Basin, adjacent to the Rajmahal Traps, also contains N–S-oriented, E-dipping step-faults that presumably formed during the pre-Rajmahal break-up of Gondwana [27,35,49]. Furthermore, there is a broad and elongated positive gravity anomaly with its axis along the western boundary, and several +10 mGal to +25 mGal gravity highs along either side of the southwestern boundary (Kathikund to Deocha) of the Rajmahal outcrop [35,36] (Figure 2). Multichannel reflection data and elastic thickness structure of the lithosphere also indicate the presence of a north–south-oriented pseudofault under the Rajmahal Traps whose formation is attributed to the activity of the Kerguelen plume [50,51]. Thus, the Rajmahal eruptions may have been guided by faults generated through reactivation of a major, pre-existing N–S-oriented mid-Neoproterozoic lineament during the break-up of Gondwana.

## 3. Sample Locations and Bulk Compositions

For the purpose of mineral thermobarometry, 40 basalt samples containing phenocrysts were analyzed from the northeastern (Sahibganj, Moti Jharna, Maharajpur, Taljhari, Tinpahar, Kherwa, Berhait), northwestern (Dariachak), and central (Litipara, Amrapara, Kundpahar) sectors of the Rajmahal outcrop (Figure 2, Table 1). The bulk compositions of four of the samples are available from [10]. According to the TAS diagram [44], samples TT3 and LH1 from near Litipara are basalts, and samples LA2 (near Litipara) and RB88-35 (Tinpahar) are basaltic andesites (Figure 3a). According to the Ti/Zr vs. Zr/Y plot [10], samples LA2, TT3, and LH1 belong to the uncontaminated Group I, and sample RB88-35 belongs to the contaminated Group II Rajmahal basalts (Figure 3b). The bulk analyses of these samples show totals higher than ~99 wt% [10], and LOI values determined by [21] for the Rajmahal basalts are <1 wt.%. This indicates that the Rajmahal magmas were essentially anhydrous.

**Table 1.** Mineral assemblages in the Rajmahal basalts.

| Region | Location | | Aug | Pgt | Pl | Hem | Ilm | Other |
|---|---|---|---|---|---|---|---|---|
| Northeast | | | | | | | | |
| | Sahebganj | RB88-12 | mph | mph | phen | | gm | |
| | Moti Jharna | RB88-19 | phen | | phen | | gm | |
| | Maharajpur | RB88-16 | phen | mph | phen | gm | gm | |
| | | RB88-17 | phen | | phen | gm | gm | |
| | | RB88-18 | phen | | phen | gm | gm | |
| | Taljhari | EB89-154 | phen | | phen | | | |
| | | RB88-24 | phen | | phen | gm | gm | |
| | Tinpahar | RB88-31 | phen | | phen | | gm | Mgh |
| | | RB88-32 | phen | | phen | gm | gm | |
| | | RB88-39 | mph | | phen | | | Ol |
| | | RB88-35 | mph | | phen | gm | gm | Cumm |
| | Kherwa | EB89-112 | mph | mph | phen | gm | | Zeol |
| | | EB89-117 | mph | mph | phen | gm | gm | Zeol |
| | | EB89-118 | mph | mph | phen | gm | | |
| | Berhait | EB89-121 | phen | mph | phen | gm | gm | |
| | | EB89-123 | mph | | phen | | gm | |
| | | EB89-127 | mph | | mph | gm | | |
| | | EB89-128 | mph | mph | phen | gm | gm | |
| Northwest | | | | | | | | |
| | Dariyachak | DAR-2GB | phen | | phen | | gm | Mgh |
| | | DAR-5YL | phen | mph | phen | gm | gm | |
| | | DAR10-90 | phen | | phen | gm | gm | |

**Table 1.** *Cont.*

| Region | Location | | Aug | Pgt | Pl | Hem | Ilm | Other |
|---|---|---|---|---|---|---|---|---|
| Central | | | | | | | | |
| | Litipara-Amrapara Rd. | LA1 | phen | | phen | gm | gm | |
| | | LA2 | phen | | phen | gm | gm | |
| | | LA4 | phen | | phen | gm | gm | |
| | | LA5 | phen | | phen | gm | gm | |
| | | LA6 | phen | | phen | gm | gm | |
| | Litipara-Simlong Rd. | TT1 | phen | | phen | gm | gm | |
| | | TT2 | phen | | phen | gm | | |
| | | TT3 | phen | mph | phen | gm | gm | |
| | | TT4 | phen | | phen | gm | gm | |
| | Litipara-Hiranpur Rd. | LH1 | phen | | phen | gm | gm | |
| | Amrapara | BL1 | phen | | phen | gm | gm | Kfs, Zeol |
| | | BL3 | phen | | phen | gm | gm | Kfs, Zeol |
| | Kundpahar | KP1 | phen | | phen | gm | gm | |
| | | KP2 | mph | | phen | gm | gm | Kfs, Zeol |
| | | KP3 | mph | | phen | gm | gm | |
| | | KP4 | mph | | phen | | | |
| | | KP5 | phen | | phen | gm | gm | |
| | | KP6 | phen | | phen | gm | gm | |
| | | KP7 | phen | | phen | gm | gm | |

Aug—augite, Pgt—pigeonite, Pl—plagioclase, Hem—hematite, Ilm—ilmenite, Mgh—maghemite, Ol—olivine, Cumm—cummingtonite, Kfs—K-feldspar, Zeol—zeolite, phen—phenocryst, mph—microphenocryst, gm—groundmass.

## 4. Analytical Methods

Textural studies and mineral analyses were performed on a JEOL JXA-8200 Superprobe electron probe microanalyzer (EPMA) at Massachusetts Institute of Technology, Cambridge, MA, USA operating with a 15 kV accelerating voltage, a 10 nA beam current, and 1 μm beam diameter. Typical counting times were 40 s per element that yielded accumulated counts with 1σ standard deviations of 0.3%–1.0% for major elements and 1%–5% for minor elements from counting statistics. The EPMA was calibrated using a set of synthetic and natural standards (DJ35 diopside-jadeite, ALP7 aluminous orthopyroxene, Synthetic anorthite, Amelia albite, Marjalahti olivine, rutile, hematite). The raw data were corrected for matrix effects with the CITZAF package [52].

## 5. Petrography and Mineral Chemistry

The samples are all of porphyritic basalt with phenocrysts (~2–4 mm) and microphenocrysts (~100–500 μm) surrounded by a fine-grained groundmass (Figure 4). The mineral assemblage in each sample is provided in Table 1. The phenocrysts are dominated by tabular clinopyroxene (augite) and lath-shaped plagioclase. Pigeonite occurs in some samples, notably from Berhait and Kherwa. Olivine is present in only one sample from Tinpahar. Rare K-feldspar occurs in one sample from Kundpahar and two samples from Amrapara. The average phenocryst content is ~5 vol.% with subequal amounts of augite and plagioclase. The groundmass consists of clinopyroxene and plagioclase. Ilmenite and hematite are ubiquitous in the groundmass and two samples contain maghemite. Some of the samples contain small spherules filled with zeolites.

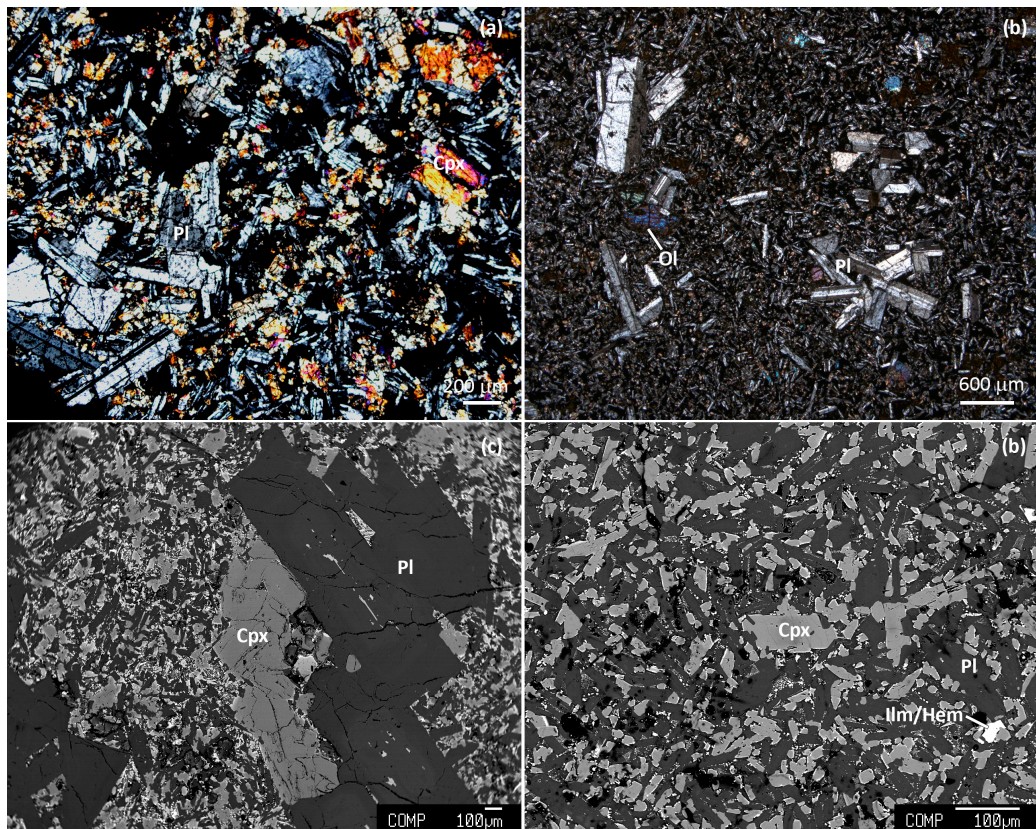

**Figure 4.** Cross-polarized light images of (**a**) Maharajpur basalt showing plagioclase (Pl) and clinopyroxene (Cpx) phenocrysts, and (**b**) Tinpahar basalt showing Pl and olivine (Ol) phenocrysts, and backscattered electron images of (**c**) Taljhari basalt showing Pl and Cpx phenocrysts, and (**d**) Kherwa basalt showing Pl and Cpx microphenocrysts surrounded by a fine-grained groundmass comprising Cpx, Pl, ilmenite (Ilm), and hematite (Hem).

The composition of the phenocrysts and microphenocrysts are summarized in Table 2 and presented in detail in Table S1. Clinopyroxene does not show compositional zoning with the exception of two samples from Dariachak in the northwest. It is dominantly augite, and its composition range in samples from the northeast, northwest and central sectors are $En_{45-56}Fs_{8-21}Wo_{29-39}$, $En_{48-54}Fs_{9-15}Wo_{36-40}$, and $En_{47-54}Fs_{7-19}Wo3_{1-41}$, respectively (Table 2). Augite from the central sector are slightly higher in the wollastonite component compared to the northeastern sector (Figure 5a). The latter is also slightly higher in jadeite, though the jadeite and aegirine contents are low ($\leq$2 mole%, Figure 5b). The composition range of pigeonite is $En_{42-74}Fs_{18-46}Wo_{4-12}$ considering all samples.

Plagioclase is mildly zoned with higher albite contents near the rim, especially in samples from the northwest and central sectors (Figure 5c). Its composition ranges from labradorite to bytownite, and rarely andesine (Table 2, $An_{57-82}Ab_{18-42}Or_{0-4}$, $An_{45-71}Ab_{28-54}Or_{0-1}$, and $An_{51-81}Ab_{19-48}Or_{0-1}$, in the northeast, northwest, and central sectors).

The olivine in a Tinpahar (northeast) sample has a composition of $Fo_{71-76}Fa_{24-30}$. The composition ranges of ilmenite and hematite are $Hem_{0-7}Ilm_{85-97}Pph_1Gk_{1-9}$, and $Hem_{43-92}Ilm_{6-53}Pph_{0-12}Gk_{0-6}$, respectively (Table 2, Figure 5d), considering all samples.

**Table 2.** Composition range of minerals in the Rajmahal basalts.

| **A. Augite and Plagioclase** | | | | | | |
|---|---|---|---|---|---|---|
| **Sector** | **Northeast** | | **Northwest** | | **Central** | |
| | core | rim | core | rim | core | rim |
| Augite | | | | | | |
| En | 45–56 | 49–52 | 48–54 | 49–50 | 47–54 | 49–54 |
| Fs | 8–21 | 10–15 | 9–12 | 11–15 | 7–15 | 7–19 |
| Wo | 29–39 | 34–39 | 36–40 | 37–40 | 36–41 | 31–41 |
| Mg# | 68–87 | 76–83 | 80–85 | 77–82 | 76–89 | 73–88 |
| Plagioclase | | | | | | |
| An | 57–82 | 58–75 | 61–71 | 45–66 | 60–81 | 51–77 |
| Ab | 18–42 | 25–41 | 28–38 | 34–54 | 19–39 | 22–48 |
| Or | 0–4 | 0–3 | 0–1 | 0–1 | 0–1 | 0–1 |

| **B. Other minerals** | | | | | | |
|---|---|---|---|---|---|---|
| **All sectors** | | | | | | |
| | Ilmenite | Hematite | | Pigeonite | | Olivine |
| Hem | 0–7 | 43–92 | En | 42–74 | Fo | 71–76 |
| Ilm | 85–97 | 6–53 | Fs | 18–46 | Fa | 24–30 |
| Pph | 1–1 | 0–12 | Wo | 4–12 | | |
| Gk | 1–9 | 0–6 | Mg# | 48–81 | | |

En—enstatite, Fs—ferrosilite, Wo—wollastonite, Mg# = 100.molar Mg/(Mg + Fe), An—anorthite, Ab—albite, Or—orthoclase, Hem—hematite, Ilm—ilmenite, Pph—pyrophanite, Gk—geikelite, Fo—forsterite, Fa—fayalite.

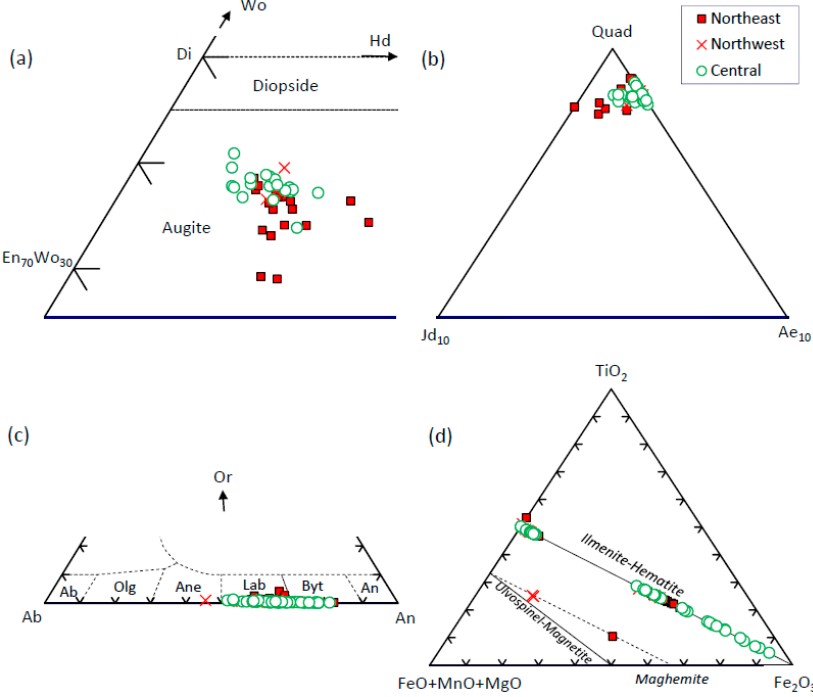

**Figure 5.** Chemical composition of minerals in the Rajmahal basalts from the northeast, northwest, and central sectors (see Figures S1–S3 for further details). (**a,b**) Average augite (En—enstatite, Di—diopside, Hd—hedenbergite, Wo—wollastonite, Quad—quadrilateral components, Jd—jadeite, and Ae—aegirine) plotted according to [53]. (**c**) Plagioclase (Ab—albite, Olg—oligoclase, Ane—andesine, Lab—labradorite, Byt—bytownite, An—anorthite, Or—orthoclase). (**d**) Oxides.

## 6. Thermobarometry

The pressure–temperature (P-T) of crystallization of the Rajmahal basalts were calculated using clinopyroxene compositions, clinopyroxene-bulk equilibria, and whole-rock (bulk) compositions.

### 6.1. Clinopyroxene Thermobarometry

This method uses only clinopyroxene composition to determine the P-T of crystallization. The T-dependent barometric expression (Equation (32a)) and P-dependent thermometric expression (Equation (32d)) of [54] were solved simultaneously to obtain P-T. These equations are based on multiple regression of clinopyroxene compositions (854 analyses for (32a), 910 analyses for (32d)) obtained from anhydrous partial melting experiments on basalts in the P-T range of 1 bar–75 kbar/800–2200 °C. The equations use the enstatite–ferrosilite and diopside–hedenbergite components and cation proportions of clinopyroxene calculated on the basis of six oxygen atoms. The quoted uncertainties are $\pm 3.1$ kbar and $\pm 58$ °C for clinopyroxene crystallizing from anhydrous melts [54]. In addition, the P-T conditions were also calculated with the random forest machine learning-based algorithms of [55,56] that provided independent estimates of the uncertainties. The two studies [55,56] use the same methodology, but [56] uses an expanded dataset that includes alkalic liquids. These thermobarometers are also based on clinopyroxene compositions obtained from partial melting experiments on basalts that cover a P-T range of 0.002–30 kbar/750–1250 °C.

Using the augite compositions in Table S1 and Equations (32a) and (32d) of [54], the P-T of clinopyroxene crystallization are 1 bar–5.4 kbar ($\pm 3.1$ kbar) and 1134–1195 °C ($\pm 58$ °C) (Table 3, negative pressure values $>-2$ kbar are considered as 1 bar). Samples from Tinpahar and Moti Jharna in the northeastern sector show the highest average pressures of 5.4 kbar and 3.2 kbar, respectively. All other samples show pressures of <2 kbar. The method of [55] also yields low pressures (2.0–4.6 kbar, highest pressures at Tinpahar and Moti Jharna) but with lower uncertainties ($\pm 0.6$–2.6 kbar) compared to [54] (Table 3). With the method of [56], the pressures are 0.0–1.8 kbar with uncertainties of less than $\pm 4.5$ kbar. The temperature estimates are similar with the methods of [55] (1093–1151 °C, $\pm 19$–62 °C) and [56] (1119–1157 °C, $\pm 21$–48 °C) compared to [54] (Table 3).

### 6.2. Clinopyroxene-Bulk Thermobarometry

The P-T conditions of crystallization were determined with the clinopyroxene-anhydrous liquid thermobarometer (Equations (P1) and (T1)) of [57] with quoted uncertainties of $\pm 1.4$ kbar and $\pm 27$ °C. The Cpx-bulk equilibrium was assessed in terms of the $Fe^{2+}$-Mg distribution, the equilibrium value of the Cpx-bulk $K_D(Fe^{2+}$-Mg) being $0.28 \pm 0.08$ [54]. A knowledge of the bulk FeO content is necessary to apply this thermobarometer. The oxidation state of the bulk can be determined from the olivine-liquid [58] and magnetite–ilmenite [59] equilibria. Equation (8) of [58] provides a method to calculate the bulk $Fe^{3+}/\Sigma Fe$ ratio from the olivine-melt $Fe^{T}$-Mg distribution (where $Fe^{T}$ is total Fe) in samples containing equilibrium olivine. The bulk $Fe^{3+}/\Sigma Fe$ ratio can also be calculated from the magnetite–ilmenite equilibrium that yields oxygen fugacity [59], which can be converted to $Fe^{3+}/\Sigma Fe$ using Equation (6b) of [60]. Unfortunately, bulk composition of the olivine-bearing sample RB88-39 is not available, and magnetite is absent in all of the samples analyzed. So, the bulk $Fe^{3+}/\Sigma Fe$ ratio could not be determined, and assumed values were used to assess Cpx-bulk equilibrium.

**Table 3.** Clinopyroxene thermobarometry of the Rajmahal basalts.

| Location | Sample | P kbar | T °C | P kbar Avg | T °C Avg | P kbar | T °C | P kbar | T °C |
|---|---|---|---|---|---|---|---|---|---|
| | | | | Equations (32a) and (32d) [54] | | [55] | | [56] | |
| **Northeast sector** | | | | | | | | | |
| Sahibganj | RB88-12 | 0.3 | 1156 | 0.3 | 1156 | 2.0 ± 1.0 | 1138 ± 44 | 0.0 ± 0.0 | 1139 ± 21 |
| Moti Jharna | RB88-19 | 3.2 | 1180 | 3.2 | 1180 | 4.6 ± 2.0 | 1151 ± 19 | 0.0 ± 2.3 | 1157 ± 25 |
| Maharajpur | RB88-16 | 0.001 | 1119 | 1.6 | 1145 | 2.0 ± 0.7 | 1133 ± 46 | 0.0 ± 0.1 | 1125 ± 23 |
| | RB88-18 | 3.2 | 1172 | | | | | | |
| Taljhari | EB89-154 | 0.001 | 1147 | 0.001 | 1147 | 2.0 ± 0.5 | 1137 ± 52 | 0.0 ± 0.2 | 1141 ± 23 |
| | RB88-24 | 0.001 | 1147 | | | | | | |
| Tinpahar | RB88-31 | 4.6 | 1209 | 5.4 | 1195 | 4.5 ± 2.2 | 1127 ± 43 | 1.8 ± 4.5 | 1147 ± 33 |
| | RB88-32 | 4.4 | 1196 | | | | | | |
| | RB88-39 | 7.5 | 1218 | | | | | | |
| | RB88-35 | 5.0 | 1156 | | | | | | |
| Kherwa | EB89-112 | 0.001 | 1111 | 1.3 | 1136 | 3.7 ± 2.1 | 1099 ± 62 | 0.0 ± 0.7 | 1119 ± 43 |
| | EB89-117 | 2.7 | 1160 | | | | | | |
| Berhait | EB89-121 | 6.6 | 1213 | 1.8 | 1160 | 3.3 ± 2.6 | 1148 ± 32 | 0.4 ± 1.0 | 1134 ± 34 |
| | EB89-123 | 0.001 | 1127 | | | | | | |
| | EB89-127 | 0.1 | 1147 | | | | | | |
| | EB89-128 | 0.5 | 1152 | | | | | | |
| **Northwest sector** | | | | | | | | | |
| Dariachak | DAR-2GB | 2.0 | 1170 | 0.7 | 1134 | 2.0 ± 0.8 | 1093 ± 42 | 0.1 ± 0.3 | 1137 ± 28 |
| | DAR-5YL | 0.001 | 1100 | | | | | | |
| | DAR10-90 | 0.001 | 1131 | | | | | | |
| **Central sector** | | | | | | | | | |
| Litipara-Amrapara | LA1 | 2.2 | 1160 | 1.5 | 1160 | 2.0 ± 0.6 | 1137 ± 50 | 0.0 ± 0.6 | 1139 ± 29 |
| | LA2 | 3.0 | 1177 | | | | | | |
| | LA4 | 0.8 | 1155 | | | | | | |
| | LA5 | 0.6 | 1156 | | | | | | |
| | LA6 | 0.8 | 1154 | | | | | | |
| Litipara-Simlong | TT1 | 2.8 | 1185 | 1.1 | 1163 | 3.1 ± 1.9 | 1141 ± 32 | 0.0 ± 0.5 | 1149 ± 25 |
| | TT2 | 0.04 | 1162 | | | | | | |
| | TT3 | 1.7 | 1166 | | | | | | |
| | TT4 | 0.001 | 1138 | | | | | | |
| Litipara-Hiranpur | LH1 | 0.1 | 1155 | 0.1 | 1155 | 4.1 ± 2.6 | 1150 ± 27 | 0.0 ± 1.0 | 1149 ± 28 |
| Amrapara | BL1 | 2.3 | 1188 | 1.9 | 1182 | 3.8 ± 2.4 | 1096 ± 51 | 0.0 ± 1.2 | 1152 ± 48 |
| | BL3 | 1.6 | 1175 | | | | | | |
| Kundpahar | KP1 | 3.6 | 1192 | 1.9 | 1162 | 2.3 ± 1.1 | 1123 ± 52 | 0.0 ± 0.5 | 1136 ± 27 |
| | KP2 | 3.3 | 1190 | | | | | | |
| | KP3 | 2.0 | 1157 | | | | | | |
| | KP4 | 0.7 | 1156 | | | | | | |
| | KP5 | 2.5 | 1161 | | | | | | |
| | KP6 | 0.001 | 1125 | | | | | | |
| | KP7 | 1.3 | 1153 | | | | | | |

Assuming a bulk $Fe^{3+}/\Sigma Fe$ ratio of 0.1, augite in samples LA2, TT3 and LH1 shows equilibrium Cpx-bulk $Fe^{2+}$-Mg distribution ($K_D(Fe^{2+}$-Mg)) values between 0.29 and 0.32, Table 4). Assuming bulk $Fe^{3+}/\Sigma Fe$ ratios of zero and 0.2 change the Cpx-bulk $K_D(Fe^{2+}$-Mg) to 0.26–0.29 and 0.32–0.35, respectively, which are also within the variability of the equilibrium value [54]. Hence, P-T of the abovementioned samples were calculated using the clinopyroxene-liquid equilibrium with the formulations of [57]. Using the augite compositions in Table S1 and bulk compositions in [10], Equations (P1) and (T1) of [57] yielded P-T of 1 bar–4.6 kbar (±1.4 kbar) and 1137–1185 °C (±27 °C) for samples LA2, TT3, and LH1. These P-T results are similar to the P-T obtained from augite composition for the same samples (0.1–3.0 kbar, 1155–1177 °C) with the formulations of [54] (Table 4).

**Table 4.** Clinopyroxene-bulk and whole rock thermobarometry of the Rajmahal basalts.

| Location | Sample | P (kbar) | T (°C) | $K_D$ [a] | Pl (%) [b] |
|---|---|---|---|---|---|
| **Equations (P1) and (T1) [57]:** | | | | | |
| Central sector | | | | | |
| Litipara-Amrapara | LA2 | 4.6 | 1185 | 0.29 | |
| Litipara-Simlong | TT3 | 1.6 | 1166 | 0.32 | |
| Litipara-Hiranpur | LH1 | 0.001 | 1137 | 0.31 | |
| **[61], and Equation (16) [54]:** | | | | | |
| Northeast sector | | | | | |
| Moti Jharna | 88-21 | 4 | 1188 | | 4 |
| Tinpahar | 88-30 | 2 | 1168 | | 7 |
| | 88-42 | 3.5 | 1183 | | 4 |
| Northwest sector | | | | | |
| S of Dariachak [c] | RJ1-25-1 | 1 | 1168 | | 6 |
| S of Dariachak [d] | RJ1-26-7 | 0.001 | 1147 | | 4 |
| Central sector | | | | | |
| Litipara-Simlong | TT3 | 2 | 1169 | | 5 |
| Litipara-Hiranpur | LH1 | 0.001 | 1158 | | 10.5 |
| W of Pakur [e] | RJ1-30-3 | 0.001 | 1158 | | 3.5 |
| | RJ1-30-4 | 0.001 | 1156 | | 3.5 |
| Kundpahar | KP6 | 0.001 | 1157 | | 10 |

[a] $Fe^{2+}$-Mg exchange coefficient for Cpx-bulk, [b] amount of plagioclase subtracted from bulk to determine melt composition, [c] Lalmatia, [d] Bejam Pahar, [e] Dhanbad village.

### 6.3. Whole Rock Thermobarometry

This method uses the predicted pressure-dependent fractionation path of a basalt to determine its pressure of crystallization. During fractionation, the basaltic melt is in equilibrium with one or several different mineral phases. At a multiple saturation point (MSP), the melt is in equilibrium with multiple phases. Parameterized expressions of experimental data [61] predict the composition of a basaltic liquid at its plagioclase lherzolite (Ol + Pl + Cpx + Opx) multiple saturation point (PL-MSP) as a function of pressure and bulk composition. The PL-MSP provides an anchor point for the fractional crystallization path that is defined by the olivine control line, and the Ol-Pl and Ol-Pl-Cpx cotectic boundaries whose intersection is determined by the method of [62]. The bulk composition of the sample and its PL-MSPs at different pressures are plotted in the Ol-Pl-Cpx (from Qz) and Ol-Cpx-Qz (from Pl) pseudoternary projections of the basalt tetrahedron according to the methods of [63,64]. If the sample shows displacement from its fractionation path, it contains excess crystal accumulation. Subtracting the excess crystals from the bulk results in a corrected bulk composition, and the sample plots on its fractionation path at a specific pressure. The corrected bulk composition represents the composition of the melt derived directly from its primitive parental magma by fractional crystallization at the specific pressure. The estimated pressure is accurate within ±2.5 kbar [61]. Using the corrected composition of the melt, its temperature is then calculated with Equation (16) of [54], which has a quoted uncertainty of ±27 °C.

The bulk compositions of the uncontaminated Rajmahal Group I basalts (excluding basaltic andesites) [10,21] and their PL-MSPs at different pressures were plotted in the Ol-Pl-Cpx (from Qz) and Ol-Cpx-Qz (from Pl) pseudoternary projections of the basalt tetrahedron (Figure 6). The basalts show displacement from their projected PL-MSPs toward the plagioclase apex in the Ol-Pl-Cpx diagram (Figure 6a), indicating cumulus enrichment of plagioclase. Subtracting 4%–10% equilibrium plagioclase from the bulk results in corrected bulk compositions of the samples that plot on their PL-MSPs between 1 bar and 4 kbar pressures in Figure 6a. Note that the Ol + Pl + Cpx cotectic boundaries shown in the Ol-Cpx-Qz diagram (Figure 6b) approximately project to the same point as the PL-MSP in Figure 6a, and it is not possible to determine whether a sample plots on its Ol + Pl + Cpx cotectic or on its PL-MSP only from Figure 6a. However, Figure 6b clearly shows that the samples plot on their Ol + Pl + Cpx cotectics at 1 bar–4 kbar (±2.5 kbar)

pressures. Note that because Figure 6b is projected from plagioclase, subtracting plagioclase from the bulk has no effect on the sample's position. The corrected bulk compositions are the compositions of the Ol-Pl-Cpx saturated melts that are derived directly from their primitive parental magmas through fractional crystallization. The temperatures of these melts calculated with Equation (16) of [54] are 1147–1188 °C (±27 °C). Notably, the calculated P-T of samples TT3, LH1, and KP6 (1 bar–2 kbar, 1157–1169 °C) are similar to the P-T determined from augite composition (1 bar–1.7 kbar, 1125–1166 °C) with the formulations of [54] (Table 4).

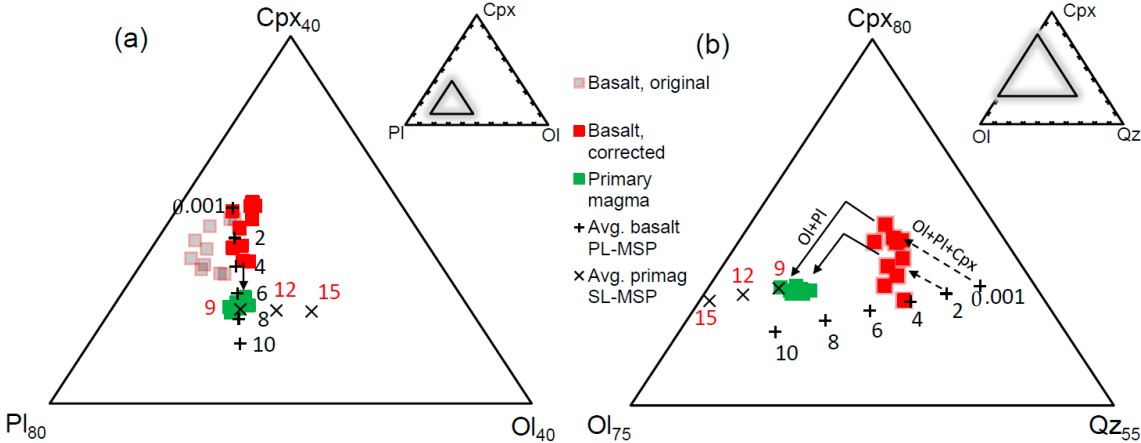

**Figure 6.** Portions of the pseudoternary projections of the basalt tetrahedron: (**a**) Ol-Pl-Cpx from Qz, and (**b**) Ol-Cpx-Qz from Pl (after [63,64]) showing the compositions of the Rajmahal Group I basalts and their corresponding melts (determined by subtracting plagioclase). For clarity, only the multiple saturation points for plagioclase lherzolite (PL-MSP at 1 bar and 2–10 kbar with 2 kbar intervals, after [59]) of the average basalt and spinel lherzolite (SL-MSP at 9–15 kbar with 3 kbar intervals, after [63]) of the average primary magma are shown. The solid lines with arrowheads represent examples of model reverse fractionation paths along the Ol + Pl + Cpx and Ol + Pl cotectic boundaries, and the dashed lines with arrowheads in (**b**) represent part of the inferred Ol + Pl + Cpx cotectic boundary at 1 bar and 2 kbar.

## 7. Primary Magma Modeling

Modeling was performed only on the Rajmahal Group I basalts [10,21], whose isotopic compositions indicate that they are uncontaminated by continental crust [10,18,21,26,31]. Because these basalts crystallized at pressures of ≤5 kbar, they evolved by fractional crystallization of olivine, followed by Ol + Pl and Ol + Pl + Cpx from their primitive parental magmas [61,65]. Hence, their primary magmas were modeled through low-pressure reverse fractional crystallization involving addition of Ol + Pl + Cpx (stage 1) and Ol + Pl (stage 2, the olivine-only stage was not necessary) to the bulk [66–69] (Figure 6). The phase assemblages were added in small steps (step size < 0.5%) with phase proportions and compositions shown in Table 5 and Table S2. Equilibrium $Fe^{2+}$-Mg distribution between olivine-liquid ($K_D(Fe^{2+}$-Mg) = 0.3, [70]) and Cpx-liquid ($K_D(Fe^{2+}$-Mg) = 0.25), and equilibrium Ca-Na distribution between plagioclase-liquid [65] were maintained at each step of the calculation. In the Ol-Cpx-Qz projection (Figure 6b), the melt moved toward the Ol-Cpx sidebar in stage 1, and toward the olivine apex in stage 2. In the Ol-Pl-Cpx projection (Figure 6a), the melt remained approximately stationary in stage 1, and moved toward the Ol-Pl sidebar in stage 2. The phase proportions and the switching point between stage 1 and stage 2 with constraints from [62] were adjusted so that the melt moved toward its spinel lherzolite MSP (SL-MSP) at high pressures predicted by parameterized expressions of experimental data [66] (uncertainties in P-T at the SL-MSP: ±1.5 kbar, ±11 °C). At the end of the calculation, the melt was in equilibrium with mantle olivine ($Fo_{90-91.5}$, for different samples), and it plotted exactly on its SL-MSP at a specific high pressure (Figure 6). The result was unique, as any deviation from the phase proportions and switching point be-

tween stages 1–2 would result in a melt not on its lherzolite MSP at any pressure, though it may show equilibrium with mantle olivine. In stage 1, MgO, Ni, CaO, and $Al_2O_3$ increased, and $SiO_2$, $FeO^T$ (total FeO), and $Na_2O$ decreased (Figure 7). In stage 2, the oxides and Ni variations followed the same trends except for CaO, which decreased slightly (Figure 7).

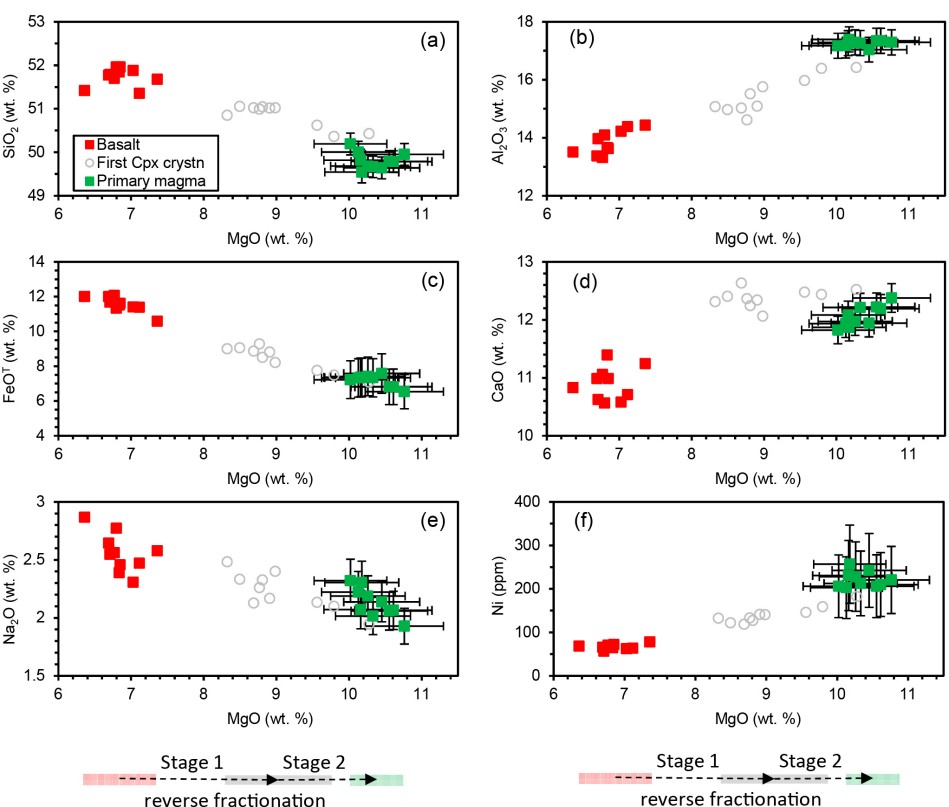

**Figure 7.** Bivariate plots showing the variation of the major oxides (**a**–**e**) and Ni with MgO (**f**) for the Rajmahal Group I basalts (corrected melt compositions) and their primary magmas. The compositions of melt in which clinopyroxene first crystallizes (end of stage 1) are also plotted. The direction of reverse fractionation with increasing MgO is shown by arrows at the bottom.

**Table 5.** P-T of crystallization and last equilibration of primary magma with mantle for the Rajmahal Group I basalts.

| | Crystallization | | | Stage 1 [a] | | | Stage 2 [a] | | | Primary Magma [b] | |
|---|---|---|---|---|---|---|---|---|---|---|---|
| | P (kbar) | T (°C) | F% | Ol | Pl | Cpx | F% | Ol | Pl | P (kbar) | T (°C) |
| Northeast sector | | | | | | | | | | | |
| 88-21 | 4 | 1188 | 60.7 | 13.7 | 52.4 | 33.9 | 12.2 | 28.6 | 71.4 | 9 | 1286 |
| 88-30 | 2 | 1168 | 60.3 | 13.0 | 52.6 | 34.4 | 19.8 | 30.2 | 69.8 | 9 | 1284 |
| 88-42 | 3.5 | 1183 | 57.8 | 13.0 | 53.4 | 33.6 | 14.8 | 30.2 | 69.8 | 9 | 1284 |
| Northwest sector | | | | | | | | | | | |
| RJ1-25-1 | 1 | 1168 | 46.8 | 12.2 | 53.1 | 34.7 | 22.9 | 30.2 | 69.8 | 9 | 1280 |
| RJ1-26-7 | 0.001 | 1147 | 57.8 | 10.9 | 52.6 | 36.4 | 30.3 | 30.2 | 69.8 | 9 | 1280 |
| Central sector | | | | | | | | | | | |
| TT3 | 2 | 1169 | 53.8 | 12.5 | 53.6 | 33.9 | 19.8 | 29.4 | 70.6 | 8 | 1268 |
| LH1 | 0.001 | 1158 | 54.2 | 13.0 | 51.9 | 35.2 | 26.0 | 28.6 | 71.4 | 8.5 | 1275 |
| RJ1-30-3 | 0.001 | 1158 | 55.6 | 12.1 | 50.2 | 37.7 | 29.6 | 30.2 | 69.8 | 9 | 1281 |
| RJ1-30-4 | 0.001 | 1156 | 56.5 | 10.9 | 52.6 | 36.4 | 29.6 | 30.2 | 69.8 | 9 | 1281 |
| KP6 | 0.001 | 1157 | 54.2 | 11.8 | 51.6 | 36.6 | 28.1 | 30.2 | 69.8 | 9 | 1282 |

[a] Percent fractionation and proportions of phases added. [b] Primary magmas (Mg# 73–76) are equilibrated with olivine $Fo_{90-91.5}$, Cpx Mg# 91.5–93, and Opx Mg# 91–92.

The calculations show that the basalts are the products of 47%–61% fractionation of their primary magmas, and the primary magmas were last equilibrated with mantle at a pressure of ~9 kbar and temperatures of 1275–1286 °C (excluding one sample, Table 5 and Table S2). The uncertainties in these results arise from the uncertainties in mineral-melt $K_D(Fe^{2+}-Mg)$ and the mantle olivine composition that may vary between $Fo_{88}$ and $Fo_{92}$. Assuming a $\pm 10\%$ uncertainty in the Ol-melt and Cpx-melt $K_D(Fe^{2+}-Mg)$, and $Fo_{88-92}$ mantle olivine compositions, the uncertainties in the MgO and FeO contents of the model primary magma are about $\pm 6\%$, and the uncertainties in P-T are $\pm 15\%$ and $\pm 3\%$ (e.g., $9.0 \pm 1.4$ kbar, $1280 \pm 40$ °C). The compositional uncertainties are depicted as error bars in Figure 7.

## 8. Discussion

The lowermost level (level 1, [3]) of the Rajmahal plumbing system is poorly constrained. It has been suggested on the basis of isotopic data that the Rajmahal basalts originated by melting of a MORB-type mantle source with only heat supplied by the Kerguelen Plume [21,31]. According to [21], viscous drag in a steady-state Kerguelen Plume conduit at the rifted eastern Indian margin may have helped in asthenospheric upwelling and melting of the MORB-type mantle. From inversion of rare-earth element data, [21] calculated a mantle potential temperature of ~1350 °C for the origin of the Rajmahal and Sylhet magmas, indicating a moderately hot mantle at the northern edge of the Kerguelen Plume. The P-T of last equilibration of the Rajmahal primary magmas with the mantle (~9 kbar and ~1280 °C, potential temperatures up to ~130 °C higher [69]) calculated in this study are consistent with the results of [21] and melting in the spinel lherzolite field concluded by [26]. However, [25,26] presented trace element and isotopic evidence to show that the source of the least contaminated Rajmahal and Sylhet basalts was the primitive Kerguelen Plume, as previously suggested by [10], and a MORB-type source was not involved. Using Nd-Sr isotopic data ($\varepsilon_{Nd(I)}$ between $-8.6$ and $+3.2$, $^{87}Sr/^{86}Sr_{(I)}$ between 0.70347 and 0.70965), [25,26] also modeled the origin of the contaminated Rajmahal–Sylhet basalts by mixing between lherzolite-derived melts and the Eastern Ghats granulites, and speculated that the basalts were contaminated through interaction and erosion of the Indian lithospheric root by the Kerguelen Plume. Considering that the axis of the Kerguelen Plume was south of the Rajmahal–Sylhet eruption site [21], the buoyant plume material may have moved laterally and upward through sublithospheric corridors [71–73] to accumulate and interact with the eastern Indian lithosphere.

Geophysical studies and primary magma modeling provide clues to the structure of levels 2 and 3 [3] of the Rajmahal plumbing system. Bouguer gravity data indicate the presence of a broad and elongated positive gravity anomaly with its axis along the western boundary of the Rajmahal Traps [35,36] (Figure 2). Three-dimensional gravity modeling and integration with seismic data delineated a high-density ($3$ g/cm$^3$) layer, 16–18 km thick under the Rajmahal Traps and ~12 km thick in the region to the south, above the 36–38 km deep regional Moho [36]. This indicates that the high-density layer under Rajmahal Traps is at lower crustal depths below an upwarped Moho that may be at a depth of ~20 km. The Rajmahal primary magmas were last in equilibrium with the mantle at a pressure of ~9 kbar that corresponds with a depth of ~33 km considering an average crustal density of $2.8$ g/cm$^3$. Thus, the Rajmahal primary magmas were last in equilibrium with the mantle within the high-density layer, which probably represents anomalous mantle within the lower crust.

The level 3 [3] of the Rajmahal plumbing system is characterized by the anomalous mantle at lower crustal depths, dikes and trans-crustal fractures/faults, and upper crustal magma staging chambers. Dikes are rare in the main Rajmahal outcrop [23,40], though NW to NNW-trending basaltic andesite dikes of similar age (118–109 Ma) to the Rajmahal basalts are common in the Raniganj–Giridih–Koderma region to the southwest and west of the Rajmahal outcrop [28]. The north–south oriented Eastern Indian Tectonic Zone (EITZ) is a trans-crustal orogen, as indicated by the high metamorphic pressures (~10 kbar)

of the granulites at the western contact of the Rajmahal Traps [46,48]. The subvertical faults associated with the EITZ, the Rajmahal and Saithia–Brahmani boundary faults along the eastern and western margins of the Rajmahal outcrop (Figure 2), and the faults in the basement of the Bengal Basin adjacent to the outcrop [27,35,49] may have acted as pathways for upward ascent of the Rajmahal primary magmas [74,75]. Thermobarometry in this study shows that the Rajmahal magmas crystallized in the upper crust. Robust estimates of the P-T conditions for magma crystallization by three different methods are between 1 bar and ~5 kbar, and 1093–1195 °C (Tables 3 and 4), indicating a maximum depth of ~19 km considering an upper crustal density of 2.7 g/cm$^3$. Crystallization probably occurred in small, upper crustal magma chambers distributed throughout the Rajmahal province [8,9]. The Rajmahal magmas differentiated through fractional crystallization and upper crustal assimilation [32] in the small, near-surface magma staging chambers before erupting on the surface.

Based on plate tectonic reconstructions [76], and analogy with basin evolution at the southwest Australian rifted margin [21,77] concluded that rifting associated with the breakup of Gondwana at the eastern Indian margin preceded or occurred synchronously with the eruption of the Rajmahal–Sylhet basalts. The recently dated 118–109 Ma old dikes of the Raniganj-Giridih-Koderma area [28] indicate that dike intrusion was synchronous with the emplacement of the Rajmahal Traps, and possibly occurred by exploiting fractures created by Gondwana breakup. Thus, plate tectonics may have played an important role in shaping the structure of the Rajmahal plumbing system [4,5].

## 9. Conclusions

Thermobarometry shows that the Rajmahal basalts crystallized in the upper crust (P-T of ≤5 kbar and 1093–1195 °C), and the Rajmahal primary magmas last equilibrated with the mantle near the Moho (P-T of ~9 kbar and ~1280 °C). A high-density layer below an upwarped Moho previously discovered through modeling of gravity data is probably anomalous mantle at shallow, lower crustal depths. The P-T results of this study complements the results of gravity modeling and provides a clearer picture of the upper levels of the Rajmahal plumbing system. It is likely that the mantle-derived primary magmas accumulated below the Moho and were subsequently transported via trans-crustal faults/fractures to the upper crust where the magma evolved by fractional crystallization in small staging chambers before erupting on the surface.

**Supplementary Materials:** The following supporting information can be downloaded at: https://www.mdpi.com/article/10.3390/min13030426/s1, Table S1. Chemical composition of minerals in the Rajmahal basalts; Table S2. Bulk composition of the Rajmahal Group I basalts, their primary magmas, and P-T of equilibration. Figure S1. (a) Core and (b) rim compositions of augite in the Rajmahal basalts (En—enstatite, Di—diopside, Hd—hedenbergite, Wo—wollastonite, Quad—quadrilateral components, Jd—jadeite, and Ae—aegirine) plotted according to [53]. Figure S2. (a) Core and (b) rim compositions of plagioclase in the Rajmahal basalts (Ab—albite, An—anorthite, Or—orthoclase). Figure S3. Chemical composition of Fe-Ti oxides in the Rajmahal basalts.

**Author Contributions:** Conceptualization, N.C.; methodology, N.C.; validation, N.C. and N.C.G.; formal analysis, N.C.; investigation, N.C. and N.C.G.; resources, N.C. and N.C.G.; data curation, N.C.; writing—original draft preparation, N.C.; writing—review and editing, N.C. and N.C.G.; visualization, N.C.; supervision, N.C. and N.C.G. All authors have read and agreed to the published version of the manuscript.

**Funding:** This research received no external funding.

**Data Availability Statement:** All data are included in the main text and online Supplementary Materials.

**Acknowledgments:** We are grateful for the constructive comments of two anonymous reviewers during the peer review process that substantially improved the presentation of the manuscript.

**Conflicts of Interest:** The authors declare no conflict of interest.

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
