# Peer review of "Thermobarometry of the Rajmahal Continental Flood Basalts and Their Primary Magmas: Implications for the Magmatic Plumbing System"

_minerals, doi:10.3390/min13030426_

Round 1

Reviewer 1 Report

The manuscript provides mineral and bulk rock geochemistry data of Rajmahal trap basalts, which are further used for P-T estimation via multiple processes. The authors further conduct melt modelling to estimate the primary magma compositions and further estimate the equilibration conditions via machine learning. Incorporating their new results along with the previous geochemical investigations and structural framework, they have tried to estimate the possible emplacement systematics of the Rajmahal Traps.

The title of the manuscript insights gravity anomaly data, however in the manuscript contribution of gravity data is absent. Although the implications drawn from the gravity data are valid, but seems to have been suggested by earlier workers. The manuscript displays new mineral chemical data and bulk rock data, which have been used for melt modelling and thermo-barometry. It seems to be the main focus of this paper. Therefore, including gravity anomaly data in the title does not seem justified.

In the introduction, the first two objectives are well defined however, the third objective i.e. “considering available geophysical data” (line 70) does not seem like a valid objective. The authors are required to either reframe it, or remove it from the objectives.

Further in Fig 2a. the exact study area needs to be demarcated by solid or dashed lines. Fig 2b, the authors might consider including the mafic dykes from Giridih and Koderma area as included in the inset. For this they may see recent publications by Srivastava and his co-workers [Srivastava et al., 2014 (JAES), 2020 (Lithsphere), 2023 (JAES)]. The fig also displays some empty circles. What are these? If they are present or previous sampling locations, their details need to be included in the figure caption.

In Section 2. The authors mention the upper sequences of Rajmahal traps to have several evidences of subaerial volcanism (line 92), but they do not describe or even give examples of any such evidence as they have given in case of the earlier sequences describing subaqueous eruptions. Did the authors observe any such evidences in field? Details of these need to be included. Further the authors have cited a dual classification suggested by earlier workers in line 95. They have also presented ME and REE plots of the two groups. My concern is the samples plotted in Fig 3, are they samples of the present study? Or do the plots also include samples from previous works. If it does, the authors need to demarcate them with a different sign. The authors directly go on to talk about the major and trace element data of the basalts, without any prior discussion about the results. They should first include a result section, where in the values of the important oxides and trace elements are discussed, and after that go on to classify their samples into groups. Details of interpretation from novel geochemical data should be discussed at a later stage in the discussion section, and not in the geochemical background. This would help them discuss their work in a systematic way.

In Line 101-102, the authors attribute Nb-Ta negative anomaly to crustal contamination. However, similar signatures will also be displayed by subduction modification of the mantle. I believe at this stage it is too premature to completely rule out the possibility of a subduction modified lithospheric mantle.

In Section 3: The authors first need to discuss the concentration of the important oxides and trace element before venturing into interpreting the data. In line 153, the authors talk about four samples from previous work whose geochemical data are already available. Were these samples also included in the geochemical plot? If yes, they need to be demarcated with a different symbol. Again, in line 136, the authors describe trace element ratios like Ti/Zr and Zr/Y, without any prior discussion of their numerical values and the spread of the data. Further, as I observe from the supplementary file, major and trace element analyses was conducted for some of the samples. The authors might consider including some of the bivariate plots, either vs MgO or trace element ratio plots two track the nature of the evolution of their samples. This way they can validate the conclusions drawn from modelling by comparing them with the observed differentiation trends. In line 139 the authors mention their samples to be essentially anhydrous. But their basis from this comment seems to be the high total values of the bulk rock analyses. This seems a bit presumptive. They need to calculate the LOI values for the analysed samples to properly constrain their anhydrous nature.

In section 4: The authors have discussed the calibration settings of their microprobe analyses. However, more details about the sample preparation techniques, which include thickness of coating, thickness of slide, and further standards that were used for the analyses need to be mentioned to check the quality of the data. Moreover, the authors have included bulk rock data in the supplementary file and also used it in their geochemical characterisations. But, not details about the bulk rock analyses are provided as such. This needs to be included in the text.

In section 5: The authors have only displayed BSE images; however, the main optical properties of the minerals are observed in photomicrographs. They should consider including some photomicrographs in PPL and crossed Nicols for better representation of the major mineral phases. In Line 179, the authors mention Plg grains to show mild zoning. As I believe this zoning was compositional, the authors should include a BSE image displaying the zoning. This would help understand the type of zoning, which can further be used to understand the equilibration and magma evolution processes. As from Table 2 it is understood that core rim analyses of the major mineral phases were conducted. However, in Fig 5 the authors have not shown no details about the samples plotted in the classification diagrams. Were they average data, or core, or rim, or even both? Proper details of these should be mentioned in the sample symbology and the figure caption, because the samples show chemical variation from core to rim, and if not carefully segregated, they might lead to grave mis-representation in the Fig 5 and subsequent analyses.

In section 6, the details of whole rock thermo-barometry seem well explained; but the mineral thermo-barometry seems confusing to the reader. As P-T estimation is a major theme of this paper, I suggest the authors should mention the whole equations used in the mineral thermo-barometry rather than just citing the parent paper (Line 192-193). Further details about source and limiting conditions that were used in the parent paper also need to be mentioned, and what is the reason behind selecting these three particular thermo-barometers? How suitable they are to be used in the present samples need to be checked and mentioned (Line 196-197). In the thermo-barometer mentioned in line 205, the estimated pressure itself is smaller than the quoted uncertainty (+- 4.5 Kbar). Is this thermo-barometer suitable for the present samples? In line 241, the authors mention multiple saturation point, which is also discussed in several places in the later sections and as I believe is a major component for the modelling. What does this MSP actually mean? And what is its significance? The authors should consider writing a few lines discussing what it actually is. In line 260 the authors subtract 4-10% normative plg. How was this particular value attained? In line 304-309, the authors discuss about reverse fractionation paths Stage 1 and Stage 2. However, in Fig 7, it is difficult to understand which process is stage 1 and which is stage 2. The authors need to properly demarcate this in the figure 7 for better comprehension. Moreover, they should add trendlines with individual R2 values to establish the strength of the fractionation trends.

In Section 8, the discussion from the P-T estimations are valid. However, the authors have discussed just their observations and cited earlier works when it comes to gravity anomaly data. They should discuss more of their original data and interpretations as it is one of the proxies for their inferences.

In Section 9, the authors talk about erosion of the lithospheric root of the eastern India by the underplating of the Kerguelen Plume (Line 397-399). However, after going through the manuscript, I could not find any novel proxies from this paper that suggest this conclusion. I agree that the earlier workers have suggested the same, but in conclusions, the authors should consider including on the novel and original finding. I therefore suggest them to remove this part from the conclusion section.

Further I've made some suggestions to tighten up and improve the language, these are not exhaustive, and a focus on this should be applied throughout the paper.

Line 46: Place “.” after Australia and replace though with Although. Join the next sentence i.e. “The eastern…” with a “,”

Line 304-306: Authors should consider rephrasing this sentence to bring out its meaning properly.

Line 307: replace Al2O3 by Al2O3

Line 380: Kent et al. : A uniform citation format should be followed through out the manuscript. Authors have made sequential numerical citation throughout the manuscript; however, at some places some papers are referred to in the format mentioned above. These should be changed to numerical format as well.

Final Comments: Overall the Manuscript requires restructuring. The authors should include a separate result section after the analytical techniques section (that should include both bulk and mineral chemical analyses), for discussing their analysed concentration of bulk and trace element data, further the interpretation from the geochemical data along with the thermo-barometry and modelling results should be systematically discussed in the discussion section. Further as some details have been provided about the field observations, the authors should consider including some field photographs describing the same. Herein I have included the major concerns observed in the different sections which need special attention. Some other suggestions are mentioned in the PDF of the manuscript, which also need to be noted.

Reviewer 2 Report

According to the title, this paper is about gravity anomalies and PT determination of the mantle source and their constraints on magma plumbing. In fact, it only delivers on the PT determination; the gravity data are hardly mentioned and are from a published paper and the magma plumbing is only concerned with a small part of the system as a whole. In my view the connection with the Kerguelen plume is speculative, especially as the authors do not present any isotope data. I am not a fan of the knd of thermobarometry presented here, but I guess that it has its place. There are large ranges of pressure calculated using e.g. the Putirka model (0 to 7.5 kb). What does this really mean? Polybaric crystallization or simply very large errors? A bit more introspection on the meaning of the ranges in P are required. Statements like 'Unfortunately, bulk composition of the olivine-bearing sample RB88-39 is not available, and magnetite is absent in all of the samples analyzed. So, the bulk Fe3+/Fe ratio could not be determined, and assumed values were used to assess Cpx-bulk equilibrium' do not inspire confidence. Why not just analyse RB88-39? Give a reason why this is not possible.

Having said the above, the manuscript is well written and does push a testable hypothesis for later authors to revise, so I recommend acceptance with some revision. I have attached a commented pdf, but the main issue (for me) is to make the title and abstract better reflect the content of the manuscript (see comments on pdf). Also, better evidence needs to be given for why some of the lavas are 'contaminated'. Negative Nb-Ta anomalies do not always indicate contamination; they could be due to the presence of a subducted component in the mantle source. 

Round 2

Reviewer 1 Report

Review of “Magmatic plumbing system of the Rajmahal continental flood basalts: Insights from gravity anomaly data and thermo-barometry” by Chatterjee et al.,

The manuscript provides mineral and bulk rock geochemistry data of Rajmahal trap basalts, which are further used for P-T estimation via multiple processes. The authors further conduct melt modelling to estimate the primary magma compositions and estimate the equilibration conditions via machine learning. Incorporating their new results along with the previous geochemical investigations and structural framework, they have tried to estimate the possible emplacement systematics of the Rajmahal Traps.

The reframed title seems a better fit for the manuscript. The authors have updated the manuscript incorporating the comments that were suggested earlier. This adds a significant amount of clarity to the work. I am happy to see that the authors have now clarified the geochemistry part. The plots are now better understood and so are their inferences and conclusions. The core rim figures in the supplementary file suggests that their values are not varying greatly and considering average values for the further processing is justified.  I have just one concern, that in line 175 the authors mention the samples are considerably anhydrous. As I mentioned in the earlier review about the LOI content. This issue does not seem addressed. As the authors mentioned the bulk rock analyses were conducted by earlier workers, was the LOI not calculated in the original papers? If yes, the authors should add it in the supplementary file along with the bulk rock data. Apart from this the rest of the manuscript seems fine to me.

Some minor comments:

Line 35: The authors have used numerical format of citation throughout the manuscript. However, herein the have mentioned Ernst et al. [3]. I think this should me made uniform and similar check should be run throughout manuscript.

Line 321: add ‘to’ before predict.

The figures however, are a bit blur and are pixelating on zooming to a certain degree. I don't know if its the quality of the figures itself, or a result of the upload process. I would recommend better quality of the plots to be uploaded. 

All in all, the updated manuscript looks fine to me. Lastly, I would suggest the authors to recheck for any grammatical and spelling mistakes throughout the manuscript.
